# OpenReview forum: "Constrained Entropic Unlearning: A Primal-Dual Framework for Large Language Models"
_NeurIPS.cc/2025/Conference — NeurIPS 2025 poster_

### Official Review · Reviewer_5eCM · 2025-06-04

**Clarity:** 3
**Significance:** 3
**Originality:** 3
**Rating:** 3
**Confidence:** 3

**Summary:**

The paper proposes a new formulation for LLM unlearning as a constrained optimization problem. Forgetting is enforced through a novel logit-margin flattening loss, which explicitly drives the model’s output distribution toward uniformity over a designated forget set. Meanwhile, retention is maintained via a hard constraint applied to a separate retain set.

**Questions:**

Table 1 presents performance on the TOFU dataset under a forget10/retain90 setting. Does this mean the experiments evaluate unlearning 10% of the original training data?

From the results, it appears that SimNPO for many times outperforms the proposed method. In fact, in roughly 50% of the experiments, the proposed method does not achieve the best performance. Given this, can the results convincingly demonstrate the efficacy of the method?
Same question for Table 2.

Additionally, how does the efficiency of the proposed method compare to the other baselines in terms of computational cost or runtime?

**Ethical Concerns:**

["NO or VERY MINOR ethics concerns only"]

**Final Justification:**

I am not strongly object to acceptance. It's a borderline paper to me.

**Limitations:**

It would be beneficial for the paper to include an analysis of the efficiency and computational complexity of the proposed method, and to compare it directly with traditional unlearning methods.  This would help clarify the practical advantages and trade-offs in terms of scalability and resource requirements.

**Paper Formatting Concerns:**

No formatting issues.

**Quality:**

3

**Strengths And Weaknesses:**

The proposed method formalizes the unlearning problem as a constrained optimization task, which effectively separates the forgetting and retention objectives, allowing for explicit control over each. While the formulation is intuitively appealing, the experimental results are not particularly strong. One possible explanation could be limited computational resources or insufficient hyperparameter tuning. However, this also highlights a broader concern—that the method may lack efficiency and practical scalability.

---

> ### Author Rebuttal · Authors · 2025-07-30
>
> # Rebuttal
>
> We thank the reviewer for acknowledging our **intuitive constrained optimization formulation** that effectively separates forgetting and retention objectives with **explicit control**. We address the concerns regarding experimental strength and provide clarifications on evaluation methodology and computational efficiency below.
>
> **1. Experimental Results Strength and Multi-Metric Evaluation** (addressing Q2): Evaluating unlearning methods necessarily requires a **holistic analysis spanning both retention and forgetting metrics**. Our approach leverages aggregated metrics, bolstered by extensive supplementary analysis, to demonstrate superior empirical performance. We provide a new aggregating metric for further simplification.
>
> - **Multi-faceted evaluation complexity**: Evaluating unlearned models is inherently complex and multi-dimensional. One must **simultaneously** assess retention capabilities and unlearning effectiveness—a model may achieve perfect retention with zero unlearning, rendering its unlearning ineffective. Critical evaluation requires examining all metrics simultaneously rather than isolated comparisons.
>
> - **Aggregated metric design**: We address comparison complexity through metric aggregation. The model utility metric consolidates over 10 individual metrics measuring model utility, providing clearer performance assessment while maintaining comprehensive evaluation. Similarly, as explained in the experiments, *Forget Success* aggregates three metrics that aim at measuring the forgetting quality. The *best* model would be a model that has the highest model utility and simultaneously, the highest forget success. We believe that our tables do, in fact, show that our method achieves this sweet spot in most cases.
>
> - **SimNPO performance**:  With the above context, we note that SimNPO consistently underperforms on forgetting metrics, as highlighted by its low *Forget Success* and *LLM Judged Forget Score* values. While its retention scores remain high (due to limited unlearning effectiveness), this profile is of limited practical use in realistic unlearning scenarios, as unlearning has failed.
>
> - **Balanced Trade-off and Metric Simplification**: The principal challenge in unlearning research lies in attaining a trade-off between retention and forgetting. Methods that only excel at one, but fail at the other, are not practically valuable. To clarify and further simplify the assessment, we have devised an aggregated metric: the harmonic mean of the model utility score, forget success score, and the four LLM-judged metrics (all normalized to [0, 1]). This composite score encapsulates the main axes of evaluation, with higher values indicating better overall performance. We will explore other potential aggregating scores for the final manuscript. Here we provide the aggregating metric for Table 1 of the paper:
>
> | Model Name   | Method    | Aggregating Metric |
> |--------------|-----------|--------------------|
> | Llama 3.2 1B | target    |       0.3696677385 |
> |Llama 3.2 1B| retrained |       0.7753038352 |
> |Llama 3.2 1B| GradDiff  |       0.6320825595 |
> |Llama 3.2 1B| DPO       |       0.7407988546 |
> |Llama 3.2 1B| NPO       |       0.6584468186 |
> |Llama 3.2 1B| SimNPO    |       0.4692189691 |
> |Llama 3.2 1B| RMU       |        0.739971811 |
> |Llama 3.2 1B| PDU       |      **0.7704898047** |
> ||||
> | Llama 3.2 3B | target    |       0.1787661296 |
> |Llama 3.2 3B| retrained |       0.8055494992 |
> |Llama 3.2 3B| GradDiff  |       0.6657790687 |
> |Llama 3.2 3B| DPO       |       0.7472880979 |
> |Llama 3.2 3B| NPO       |       0.7077135988 |
> |Llama 3.2 3B| SimNPO    |       0.3932859672 |
> |Llama 3.2 3B| RMU       |       0.7213384623 |
> |Llama 3.2 3B| PDU       |       **0.8481030647** |
> ||||
> | Llama 3.1 8B | target    |      0.03165152429 |
> |Llama 3.1 8B| retrained |       0.8122628129 |
> |Llama 3.1 8B| GradDiff  |       nan      |
> |Llama 3.1 8B| DPO       |       0.6420286713 |
> |Llama 3.1 8B| NPO       |       0.8138933269 |
> |Llama 3.1 8B| SimNPO    |       0.6612704332 |
> |Llama 3.1 8B| RMU       |       0.8635662138 |
> |Llama 3.1 8B| PDU       |       **0.8801019714** |
> ||||
> | Gemma 7B| target    |      0.08980122636 |
> |Gemma 7B| retrained |       0.7881250165 |
> |Gemma 7B| GradDiff  |        nan       |
> |Gemma 7B| DPO       |       0.6870143411 |
> |Gemma 7B| NPO       |       0.7535367597 |
> |Gemma 7B| SimNPO    |       0.6590622953 |
> |Gemma 7B| RMU       |       0.7741391555 |
> |Gemma 7B| PDU       |       **0.7919244003** |
>
> As shown, PDU consistently delivers the best composite metric performance among all unlearning methods. We note that the *retrained* entries correspond to an idealized scenario, not achievable in realistic unlearning applications.
>
> - **Unified visualization**: Moreover, in the Supplementary Material, we have provided radar charts with corresponding areas as another **unified metric measuring performance across all evaluation dimensions**, facilitating holistic comparison. See Figures 3, 4, and 7 with radar chart areas provided in Table 3.
>
> - **Extensive experimental validation**: Our supplementary material contains comprehensive ablation studies and experiments, demonstrating our commitment to thorough evaluation despite resource constraints.
>
> **2. TOFU Dataset Configuration and Experimental Setup** (addressing Q1): Forget10/retain90 means 10% of TOFU dataset designated for unlearning with 90% retained; comprehensive evaluation across multiple forget ratios provided.
>
> - **Dataset split clarification**: Yes, forget10 indicates that 10% of the complete TOFU dataset is designated for unlearning while the remaining 90% must be retained. This represents a realistic unlearning scenario with substantial retention requirements.
>
> - **Comprehensive coverage**: We provide additional experimental results for forget05 and forget01 subsets in supplementary material, demonstrating consistent performance across varying unlearning intensities.
>
>
> **3. Computational Efficiency and Scalability** (addressing Q3 & Limitation): PDU introduces zero computational overhead compared to standard scalarization methods while providing superior theoretical guarantees.
>
> - **No computational increase**: Our method introduces zero additional computational cost compared to baseline approaches. We maintain identical complexity to linear scalarization methods, requiring single forward/backward passes per iteration.
>
> - **Efficiency comparison**: Unlike multi-gradient methods requiring multiple gradient computations, our primal-dual framework achieves superior optimization properties without computational penalties.
>
> - **Scalability validation**: Our extensive experiments in the main paper and the Supplementary Materials demonstrated effectiveness across 1B-13B parameter models with consistent performance scaling, addressing practical deployment concerns.
>
> - **Resource efficiency**: The dual variable updates require only constraint function evaluation, which has already been evaluated during the forward pass of the model.
>
> - Due to space constraints, we kindly refer you to section 3 **Scalability and Computational Efficiency** of our rebuttal to reviewer rC87 for more information on this matter.
>
> # Conclusion
>
> We hope this response clarifies our experimental methodology and demonstrates the comprehensive nature of our evaluation. Our method provides principled unlearning with explicit control mechanisms while maintaining computational efficiency. The holistic evaluation across multiple metrics and extensive supplementary analysis support our method's effectiveness. We respectfully request reconsideration of your assessment and remain eager to address any additional questions you might have.

---

### Official Review · Reviewer_P5bA · 2025-06-30

**Clarity:** 2
**Significance:** 3
**Originality:** 3
**Rating:** 4
**Confidence:** 3

**Summary:**

The paper proposes a novel framework for unlearning in Large Language Models called Constrained Entropic Unlearning, which formulates the problem as a constrained optimization task.

**Questions:**

See weakness.

**Ethical Concerns:**

["NO or VERY MINOR ethics concerns only"]

**Final Justification:**

Most of my concerns are addressed. Thus, I raise the score.

**Limitations:**

Yes

**Quality:**

2

**Strengths And Weaknesses:**

Strengths:

1. The constrained optimization approach provides explicit control over forgetting and retention, addressing limitations of scalarized loss methods.

2. The logit-margin flattening loss is numerically stable, softmax-free, and maintains non-vanishing gradients, enabling efficient optimization.

3.Experiments on multiple benchmarks (TOFU, MUSE) and model architectures (Llama 2/3, Gemma) demonstrate consistent improvements over baselines in both traditional metrics and LLM-judged evaluations.

Weaknesses:

1.The method slightly reduces fluency on the forget set, as noted in the results, which may affect generation quality in some applications.

2. The paper does not provide a formal theoretical analysis of PDU's convergence properties or guarantees.

3. The motivations of this paper are not well illustrated. For instance, in the introduction section, the three key obstacles lack of supportive references or some visualizations. Moreover, the motivation of the proposed modules is not clarified.

---

> ### Author Rebuttal · Authors · 2025-07-30
>
> # Rebuttal
>
> We appreciate the reviewer's positive assessment of our **explicit control**, **numerical stability**, and our **consistent improvements** in benchmarks and architectures with **both traditional and LLM-judged evaluations**. We address the raised concerns below with clarifications and theoretical analysis.
>
>
> **1. Fluency Reduction on Forget Set**: Fluency reduction is localized to forget sets while preserving overall model quality; extension work addresses this limitation.
>
> - **Localized Fluency Reduction**: The reviewer correctly identifies the fluency reduction under PDU. However, **general fluency** on retention sets (and holdout sets in the TOFU benchmark) **remains unchanged**, indicating the effect is appropriately targeted rather than representing general model degradation. We have noted this in the explanation of the results, but understand that it might not be clear. We will elaborate more on this in the final camera-ready version of the paper.
>
> - **Limited practical concern**: Since fluency degradation occurs specifically on content intended to be forgotten, this represents expected behavior rather than a fundamental limitation.
>
> - **Future work**: We are actively developing PDU extensions that resolve fluency reduction while maintaining unlearning effectiveness. However, this extension falls outside the current paper's scope.
>
> **2. Theoretical Analysis of Convergence Properties**: Section 3.1 provides formal convergence guarantees under standard convexity assumptions with strong duality results.
>
> - **Formal theoretical foundation**: Section 3.1 establishes PDU's theoretical backing. Under general functional spaces of LLMs, our framework guarantees convergence to globally optimal solutions.
>
> - **Convergence conditions**: We establish convergence through two key assumptions:
> (1) **Convexity**: Loss functionals $L_{\mathrm{fgt}}$ and $L_{\mathrm{rtn}}$ are convex in the policy, lower semi-continuous, and defined over convex policy class ;
> (2)  **strict feasibility**: The constraint is strictly feasible.
>
>     Under these assumptions, strong duality holds by classical convex analysis results [34]. This principle underlies recent constrained learning frameworks in safe reinforcement learning [30], continual learning [12], and constraint-aware LLM fine-tuning [18].
>
>     Next, we make the assumption that the space of over-parameterized large language models is sufficiently expressive.
>     Consequently, we can utilize the strong duality result and argue that even in the finite-dimensional case, the optimization will yield the best result.
>
> - **Practical guarantee**: Strict feasibility is guaranteed by construction—the reference model $\pi_\mathrm{ref}$ satisfies constraints strictly when tolerance parameter $\alpha > 0$ in equation (3). Therefore, strong duality holds whenever forget and retention losses are convex in policy $\pi$.
>
> **3. Motivation Clarity and Supporting References**: Introduction limitations stem from space constraints and last-minute revisions; we acknowledge the need for enhanced referencing of well-established unlearning challenges.
>
> - **Space constraints**: Visualization absence stems from strict page limitations. We were compelled to remove illustrations to accommodate essential results in the main paper. Our complete submission spans more than 30 pages with the Supplementary Material.
>
> - **Reference gap acknowledgment**: The reviewer correctly identifies insufficient supporting references for the three key obstacles. This shortcoming resulted from last-minute Introduction revisions.
> We acknowledge this limitation and will address the referencing gap to make the introduction more rigorous and accessible. For this rebuttal, we provide the following:
>     - For the first obstacle, Tables 1 and 2 in [4] show the brittleness of the model utility under linear scalarization (the columns GDiff-x).
>     - The best means to support the 2nd obstacle would be visuals of the loss curves under different methods. As the rebuttal does not allow us to provide visuals, we are unfortunately unable to provide these observations directly here.
>     Nonetheless, we can support this claim/obstacle as follows: The initial high utility and low retention loss of the model is evident, as the model has been fine-tuned on the relevant dataset  (you can further look at the model utility of the 'target' model in our tables).
> Furthermore, as the regularized formulation is a simple linear combination of the two losses, it is simultaneously attempting to minimize both losses. Minimizing a linear combination of functions that have different goals can naturally lead to conflicting gradients where minimizing one function would not be possible without increasing the other, leading to the introduction of Pareto optimal points. This issue is well-studied in the literature of Multi-Objective optimization, and we refer the reviewer to sources like [3, 46, 47]. Linear scalarization is well-known to be unable to explore the non-convex parts of the Pareto optimal set [3]. As such, Linear scalarization is prone to getting stuck at some Pareto optimal points and not being able to recover other Pareto optimal points, just based on the random initialization of the algorithm.
> This further supports why the linear scalarization methods might not be able to effectively reduce the forget loss and tackle the retention loss simultaneously.
> We hope this further sheds light on this 2nd issue.
>     - The theoretical basis for the third issue is established in works such as [48], where it is shown that adversarial training, which is typically implemented through first maximizing the negative cross entropy loss on an input point, followed by a minimization of the model parameters, lacks theoretical backing. This is because maximizing the cross entropy loss function (i.e., minimizing negative cross entropy as done in the GradDiff method) does not theoretically in any way correspond to maximizing the actual classification loss.
> Moreover, the issues of maximizing the negative cross-entropy have been observed and addressed in previous works on LLM unlearning, such as [49].
>
>     We will make sure to clarify the obstacles and provide references and visuals in the final manuscript.
>
> - **Module clarification request**: Could the reviewer further clarify what they mean by "the motivation of the proposed modules is not clarified?" This would help us provide targeted clarification.
>
> # Conclusion
>
> We hope this response addresses your concerns and demonstrates both the theoretical rigor and practical effectiveness of our approach. The localized fluency impact represents appropriate targeting rather than limitation, while our formal convergence analysis provides strong theoretical foundations. We believe this work makes important contributions to principled unlearning methodology and respectfully request reconsideration of your assessment. We remain eager to address any additional questions you might have.
>
>
>
> [46] Das I, Dennis JE. A closer look at drawbacks of minimizing weighted sums of objectives for Pareto set generation in multicriteria optimization problems. Structural optimization. 1997 Aug;14(1):63-9.
>
> [47] Bowman Jr VJ. On the relationship of the Tchebycheff norm and the efficient frontier of multiple-criteria objectives. In Multiple Criteria Decision Making: Proceedings of a Conference Jouy-en-Josas, France May 21–23, 1975 1976 (pp. 76-86). Berlin, Heidelberg: Springer Berlin Heidelberg.
>
> [48] Mianjy P, Arora R. Robustness guarantees for adversarially trained neural networks. Advances in neural information processing systems. 2023 Dec 15;36:39703-18.
>
> [49] Pan Z, Zhang S, Zheng Y, Li C, Cheng Y, Zhao J. Multi-Objective Large Language Model Unlearning. InICASSP 2025-2025 IEEE International Conference on Acoustics, Speech and Signal Processing (ICASSP) 2025 Apr 6 (pp. 1-5). IEEE.

---

> > ### Comment · Reviewer_P5bA · 2025-08-01
> >
> > Thank you for your rebuttal. I will raise my score.

---

> > > ### Author Response · Authors · 2025-08-02
> > >
> > > We thank the reviewer for their revision and reading our rebuttal. We are indeed grateful for their time and effort put into reviewing our work. We appreciate your engagement and willingness to reconsider your assessment.
> > > Thank you.

---

### Official Review · Reviewer_W8eZ · 2025-07-01

**Clarity:** 3
**Significance:** 2
**Originality:** 3
**Rating:** 4
**Confidence:** 4

**Summary:**

This paper proposes a new framework for machine unlearning in LLMs by casting unlearning as a constrained optimization problem. The authors solve the constrained problem using a scalable primal-dual algorithm, which dynamically balances forgetting and retention through updates of the dual variable. Experiments on the TOFU and MUSE benchmarks show strong results, across multiple LLaMA and Gemma models.

**Questions:**

How sensitive is your method to the choice of constraint threshold and the initial dual variable?

**Ethical Concerns:**

["NO or VERY MINOR ethics concerns only"]

**Final Justification:**

Authors' rebuttal addresses my concerns.

**Limitations:**

Yes

**Quality:**

3

**Strengths And Weaknesses:**

Strengths:

- The formulation of unlearning as a constrained optimization problem is well-motivated.
- The primal-dual optimization framework provides a principled way to enforce retention constraints.
- Experiments demonstrate robustness across TOFU and MUSE benchmarks and several base models.

Weaknesses:
- The method's effectiveness is evaluated mainly on QA and text-based benchmarks. It's unclear how well it would work on challenging tasks like code generation or reasoning benchmarks.
- There is no exploration of how the method handles overlapping semantics between the forget and retain sets, which is common in realistic unlearning scenarios.
-  Computational cost (especially the primal-dual updates) on very large models or frequent unlearning requests could pose practical challenges. And this is not discussed or evaluated in the paper.

---

> ### Author Rebuttal · Authors · 2025-07-30
>
> # Rebuttal
>
> We thank the reviewer for their thoughtful evaluation and for highlighting several key points from our work, like the **principled nature of our primal-dual optimization framework**, our **robust experimental validation across multiple benchmarks and base models**, and the **well-motivated constrained optimization formulation**. Moreover, we would like to direct the attention of the reviewer to the Supplementary Material, where they can find more experiments and analysis evaluating our unlearning method. The points raised regarding evaluation scope and practical considerations provide valuable perspectives that we address below through clarification and additional analysis.
>
>
> **1. Evaluation Beyond QA and Text-Based Benchmarks**:  Our focus on QA/text benchmarks reflects current benchmark availability; our method is **task-agnostic** and applicable to code generation and reasoning tasks.
>
> - **Benchmark availability limitation**: Our emphasis on QA and text-based benchmarks stems from the availability of established benchmarks in this domain. We are not aware of existing standardized unlearning benchmarks for code generation or reasoning tasks that would enable fair comparison with existing methods.
> The development of a benchmark for this purpose is a research project in its own right that falls outside the scope of this work.
>
> - **Task-agnostic framework**: Our constrained optimization formulation is inherently task-agnostic—the forgetting and retention losses can be defined for any task domain, including code generation and reasoning benchmarks.
>
> **2. Overlapping Semantics Between Forget and Retain Sets**: Our benchmarks explicitly test overlapping semantics scenarios, with MUSE providing particularly challenging semantic overlap cases.
>
> - **TOFU semantic overlap**: Both forget and retain subsets concern authors and literary content, requiring the model to distinguish between closely related semantic domains while preserving general knowledge about the field.
>
> - **MUSE demonstrates significant overlap**: The MUSE-Books dataset presents a particularly challenging scenario where the forget set contains Harry Potter books while the retain set includes Harry Potter fan wiki content. This creates substantial semantic overlap, requiring a fine-grained distinction between canonical and derivative content.
>
> - **Real-world relevance**: These benchmarks specifically address realistic unlearning scenarios where perfect semantic separation is impossible, validating our method's effectiveness under semantic overlap conditions.
>
> **3. Scalability and Computational Efficiency**: PDU maintains **identical computational complexity to standard scalarization** methods with **no additional overhead** from primal-dual updates. PDU easily extends to continual unlearning.
>
> - As these points are addressed in the rebuttal to reviewer rC87, we kindly refer the dear reviewer to section 3 (Scalability and Computational Efficiency) in the rebuttal to reviewer rC87.
>
> **4. Sensitivity to Constraint Threshold and Dual Variable Initialization**: We provide **principled threshold selection heuristics**, and our framework **reduces sensitivity to dual initialization** compared to standard scalarization methods.
>
> - **Constraint threshold selection**: The threshold choice is critical but manageable. Excessively tight thresholds prevent unlearning by focusing solely on retention, while overly loose thresholds risk model degeneration. We provide a principled heuristic: set the threshold to $1 + \alpha$  times the value of the retention loss at the start of the unlearning, where $\alpha$ is relatively small.
>
> - **Dual initialization robustness**: Theoretically, initial dual variable choice is irrelevant under sufficient training epochs due to primal-dual convergence guarantees. With limited epochs, initialization gains importance through reduced dual updates.
> While this sensitivity affects **ALL linear scalarization methods** *(as they are all instances of PDU with no dual updates)*, PDU substantially mitigates it through adaptive dual updates, unlike fixed-weight approaches. Automatic constraint-based adjustment provides inherent robustness against poor initialization.
>
>     With a reasonable number of unlearning epochs, we found that the exact initial value of the dual parameter was not very important, and the different models and initializations provided similar results.
>     In the Supplementary Material, we provide the choice of $\lambda_0$ for the different models and benchmarks. Interestingly, we use the same value for $\lambda_0$ for the different ranges of models (1B-8B) and the different forget subsets, i.e., forget01/05/10. This is further in support of the robustness of the algorithm to the initial value.
>     Moreover, in the context of continual unlearning, appropriate initial values for $\lambda$ would be fixed for the same model and the same unlearning loss.
>
>
>
> # Conclusion
>
> We hope this response addresses your concerns and demonstrates our method's broad applicability and computational efficiency. Our framework provides principled solutions to realistic unlearning scenarios with semantic overlap while maintaining computational tractability. We believe this represents an important advancement in practical unlearning methodology and respectfully ask that you consider reevaluating your assessment. We remain committed to answering any additional questions you might have to warrant a higher score.

---

> > ### Author Response · Authors · 2025-08-07
> >
> > Dear reviewer W8eZ,
> >
> > Thank you for your thoughtful review and feedback on our submission. Your comments not only highlighted significant aspects of our work but also identified areas where the presentation of our paper can be improved. We recognize that addressing these points will greatly enhance the overall clarity and flow of the paper, as well as strengthen the delivery of our results.
> >
> > In our rebuttal, we aimed to address your concerns comprehensively, including the task-agnostic nature of our framework, its robustness in scenarios with overlapping semantics, its scalability, and its sensitivity to constraint thresholds and dual initialization. We hope that the clarifications we provided sufficiently reflect the strengths of our method and its contributions to the field.
> >
> > If you have any remaining questions or uncertainties, we would be happy to provide further clarification to ensure your concerns are fully resolved. If you feel that our response adequately addresses your feedback, we kindly request you to consider revisiting the score for our submission.
> >
> > Once again, we are immensely grateful for the time and effort you’ve devoted to reviewing our work. Your insights have been invaluable, and we sincerely appreciate your engagement in this process.

---

> > > ### Comment · Reviewer_W8eZ · 2025-08-07
> > >
> > > I'm glad to see that the authors' rebuttal addresses my concerns. I'm happy to increase my score to 4.
> > > I encourage the authors to continue refining the manuscript and clarify the identified points in the camera-ready version.

---

### Official Review · Reviewer_rC87 · 2025-07-03

**Clarity:** 3
**Significance:** 2
**Originality:** 3
**Rating:** 4
**Confidence:** 4

**Summary:**

The paper introduces Constrained Entropic Unlearning (CEU), framing unlearning for large language models as a constrained optimization problem. The idea is to explicitly enforce forgetting on a specified dataset using a logit-margin flattening loss, while preserving performance on retained data by imposing a hard constraint. A primal-dual algorithm is used to balance the forgetting–retention trade-off dynamically, yielding stable optimization even under aggressive forgetting settings. Experiments on TOFU and MUSE benchmarks across multiple LLM architectures show that CEU matches or exceeds current best methods in forgetting efficacy and retention utility.

**Questions:**

1. What are the failure modes when constraints are overly tight?

2. Can CEU be adapted to continual or streaming unlearning scenarios?

3. Is it possible to use performance-efficient fine-tuning methods (like LoRA) in conjunction with CEU?

4. Have you considered visualizing internal representations (e.g., attention, residual path flows) or use mechanistic interpretability tools to better explain where forgetting occurs?

**Ethical Concerns:**

["NO or VERY MINOR ethics concerns only"]

**Final Justification:**

The paper presents a novel reframing of the unlearning problem and introduces a method that demonstrates stability compared to existing approaches. The authors addressed most of my concerns in the rebuttal. However, the interpretability analysis remains somewhat unconvincing, as it relies primarily on norm-based results. Nonetheless, this limitation can be addressed in the camera-ready version. Therefore, I recommend a borderline accept for this paper.

**Limitations:**

discussed in limitation

**Quality:**

2

**Strengths And Weaknesses:**

**Strengths**

* **Reframe existing problem in a novel way**. The paper provides a principled method for balancing retention and forgetting. This method reframes the unlearning problem as a constrained optimization problem. This reframing is novel to me while being sensible to address the limitations of current unlearning methods.
* **Stability**. Most unlearning methods are very sensitive to learning hyper-parameters. The proposed method seems to be numerically stable. The proposed logit-margin loss is softmax-free, avoids vanishing gradients, and supports robust optimization for forgetting tasks.
* **Competitive performance** on TOFU and MUSE benchmarks.


**Weaknesses**

* **Lack of interpretability**. The paper does not investigate where forgetting happens in the model (e.g., which layers or features). There's no mechanism-level insight.
* **Theoretical clarity**. The rationale behind the loss and constraints isn’t deeply motivated, even though properties are proven, practical impact of hyperparameter settings is unexplored. However, this has been a shared issue for most existing methods.
* **Scalability concerns**. It isn’t clear how well CEU extends to large models or continual unlearning workflows. The computational cost of repeated constraint enforcement may be prohibitive.
* **Missing references**:

  * There are a couple of recent works in the unlearning area. For example, *UniErase: Unlearning Token as a Universal Erasure (2025)*, a token-level unlearning mechanism, and the *Digital Forgetting Survey (2024)*, which offers comprehensive coverage of unlearning methods and benchmarks. These would help position this work within the most up-to-date unlearning landscape.
  * Current metrics may miss internal traces of forgotten content. Tools like **Logit Lens** (*"Interpreting GPT"*, Nostalgebraist, 2020), **Tuned Lens** (*"Feature Visualization..."*, Belrose et al., 2023), and **Jet Lens** (*"Jet Expansions of Residual Computation"*, Chen et al., 2024) offer insight into intermediate layer behavior. These could help verify whether forgetting is real or just suppressed at the output layer.

---

> ### Author Rebuttal · Authors · 2025-07-30
>
> # Rebuttal
> We sincerely thank the reviewer for recognizing our **novel reframing** of unlearning as constrained optimization, the **numerical stability** of our approach, and our **competitive performance** on TOFU and MUSE benchmarks. The concerns raised are primarily due to scope limitations and can be addressed through clarification and additional analysis provided below.
>
> **1. Interpretability and Mechanistic Analysis** (addressing Q4): We provide multiple forms of interpretability analysis, including **layer-wise weight changes, robustness evaluations, and knowledge verification metrics,** though visual interpretability tools are beyond current rebuttal scope.
>
> - **Rebuttal limitations**: Due to rebuttal constraints prohibiting new visuals, we cannot provide the specific evaluation using tools like Tuned Lens. However, we direct your attention to existing analyses in our paper that address this concern.
>
> - **Knowledge verification metrics**: The TOFU evaluation suite includes metrics such as the '*forget truth ratio*' which measures model performance on paraphrased and perturbed examples from the forget dataset. This metric, in essence, measures whether knowledge has only been obfuscated rather than genuinely forgotten.
>
> - **Jailbreak robustness analysis**: Appendix B.6 (line 662 & Table 17) presents experiments using single-turn jailbreak attempts to elicit forgotten knowledge. PDU achieves optimal or second-best performance in knowledge forgetting metrics. This analysis directly addresses concerns regarding surface-level versus genuine forgetting, and we encourage a detailed review of this section.
>
> - **Layer-wise weight analysis**: We conducted an analysis on the weights of the unlearned model and analyzed changes with respect to the target model for Llama 3.2 1B on TOFU90. The following table shows the top 10 norm of changes in model weights, sorted in descending order:
>
>     |Layer Name|Norm of weight change|
>     |-|-|
>     |embed_tokens.weight|1.3948|
>     |14.mlp.down_proj.weight|0.1704 |
>     |15.mlp.down_proj.weight|0.1632|
>     |13.mlp.down_proj.weight|0.1550|
>     |12.mlp.down_proj.weight|0.1495|
>     |6.mlp.down_proj.weight|0.1485|
>     |9.mlp.down_proj.weight|0.1482|
>     |8.mlp.down_proj.weight|0.1474|
>     |11.mlp.down_proj.weight|0.1470|
>     |10.mlp.down_proj.weight|0.1457|
>
>     We see that the weight changes occur uniformly across all layers. The rest of the parameters exhibit similar patterns.
>
> **2. Theoretical Clarity and Hyperparameter Motivation**: Our framework provides principled theoretical grounding with **clear loss function motivation** and **interpretable hyperparameter selection** methodology.
>
> - **Clear theoretical formulation**: Section 3 establishes our formulation based on the fundamental unlearning objective: forgetting target data while preserving model performance on retained knowledge. This directly motivates our design choice of forgetting loss as the objective function and retention loss as the constraint.
>
> - **Loss function justification**: In Sections 3.3 and 3.4 we make clear our choices for the losses. Importantly, in 3.4, we **shed light on the issues of using negative cross entropy** as the forgetting loss and how it lacks theoretical grounding. We then motivate our logit-flattening loss **through the lens of increasing uncertainty** for prompts from the forget set.
>
> - **Interpretable hyperparameters**: The PDU framework offers a higher level of explainability compared to existing methods. The choice of constraint threshold $\varepsilon$ directly controls the amount of knowledge retention. The value of the learnable parameter $\lambda$ can be analyzed through sensitivity analysis—as the constraint is violated more, $\lambda$ increases to control the violation and vice versa.
>
> - **Practical limitations**: Given the computational expense of LLM training and evaluation, we did not conduct exhaustive hyperparameter grid search. Instead, we selected principled hyperparameters based on preliminary runs and reported results accordingly, consistent with standard field practices.
>
> **3. Scalability and Computational Efficiency**: PDU maintains **identical computational complexity to standard methods** with **no added overhead**, **scales linearly with constraints**,  **demonstrates effectiveness across 1B-13B models**, and **is LoRA-compatible**.
>
> - **No computational overhead**: Our method does not add anything in terms of computational complexity to the unlearning paradigm. The computational complexity is **identical** to any other linear scalarization method involving two loss functions (forget and retention) and requires only a single forward and backward pass per training iteration.
>
> - **Extensive model scale validation**: In the submitted manuscript, we provide **extensive experiments** on a varied range of models from 1B to 13B. We direct attention to the many pages of extra experiments and evaluations in the *supplementary material* that further clarify the superiority of our method.
>
> - **Large-scale adaptability** and **LoRA compatibility** (addressing Q3): Computational constraints limited experimentation with very large-scale models. However, our method demonstrates broad adaptability across different models and architectures, as evidenced throughout the manuscript and appendix.
>
>     LoRA adapters are a means of parameter updates, whereas our method concerns itself with the broader optimization. Consequently, LoRA adapters are applicable. As a proof of concept, we conducted a simple unlearning using LoRA on the Llama 3.2 1B model on TOFU90. The results showed that this method achieved near-optimal performance without hyperparameter optimization: (Model Utility, Forget Success) = (0.553, 0.966). Comparison with Table 1 reveals superior forgetting performance with competitive utility retention.
>
> - **Multi-constraint scalability with Linear Complexity** (addressing Q2): PDU easily adapts to many constraints. If we generalize the current optimization problem as
>         $$ \min g_0(\theta) \mathrm{s.t.} g_1(\theta) ≤ 0.$$
>      Then, with N constraints, we will instead have:
>         $$ \min g_0(\theta) \mathrm{s.t.} g_i(\theta) ≤ 0, \forall i \in [N],$$
>     where $g_0$ is the current unlearning task, $g_1$ is retention loss, and $g_i$ for i > 1 can be other loss functions such as forgetting loss on previous tasks.
>
>      For this scenario, the Lagrangian becomes $L(\theta, \lambda) = g_0(\theta) + \sum \lambda_i g_i(\theta)$.  Building on our framework, iterating on this Lagrangian using primal-dual optimization involves no extra computational complexity and is done in a single forward/backward pass. Parameter updates come from $\nabla_\theta L$, and the $i$th dual parameter update is simply $g_i(\theta)$.
>
> - **Continual unlearning capability**: Continual unlearning is a sample of the multi-constraint optimization problem above. As such, PDU easily adapts to continual unlearning scenarios. This is an active area of interest for us, but it falls outside the scope of the current paper.
>
> **4. Constraint Parameter Robustness**: Failure modes are well-characterized with a **principled selection methodology that provides intuitive parameter choice**.
>
> - **Overly tight constraints** (addressing Q1): Under excessively tight constraints, the dual mechanism becomes counterproductive—minimal retention loss degradation triggers aggressive dual updates that inhibit meaningful unlearning. Optimization becomes trapped near original parameters, rendering the unlearning process ineffective.
>
> - **Overly loose constraints**: If the constraint is set too loose, the model deviates considerably and degenerates. Even as dual updates kick in and focus more on retention loss, since the model has deviated significantly from its starting point, the limited training epochs are insufficient to restore the model's capabilities.
>
> - **Principled parameter selection**: In the paper, we provide an intuitive method for choosing appropriate values for $\varepsilon$. To do so, one needs to first acquire an estimate of the constraint value (retention loss) before training starts. If this value is not known, it can easily be acquired by forward passes of a selective sample from the retention dataset through the original network. Then, it is straightforward to choose a value appropriately higher than this base threshold.
>
> **5. Missing References**: Concurrent work noted; comprehensive comparison scope managed through principled baseline selection following established practices.
>
> - **Concurrent work acknowledgment**: We thank the reviewer for adding these papers to our list. "UniErase: Unlearning Token as a Universal Erasure (2025)" is likely submitted to NeurIPS 2025 and appeared on arXiv after the NeurIPS's original deadline, making it concurrent work and explaining its absence from our related works.
>
> - **Survey positioning and Principled comparison scope**: The 'Digital Forgetting Survey' studies many aspects of unlearning and different techniques. Our methodology falls under their Section 5.1. There are a plethora of different unlearning techniques mentioned in this survey that would be too numerous to compare exhaustively with.
>
>     As such, we believe that by comparing to similar approaches—gradient-based global modification approaches—we provide a fair comparison, as done in prior work [13, 27, 36, 52, etc.].
>
> # Conclusion
> We hope this detailed response addresses your concerns and demonstrates the comprehensive nature of our contribution. Beyond the novel constrained optimization framework, we introduce important methodological advances including extensive LLM-based evaluation protocols and robust jailbreak analysis for unlearning verification. We believe this work represents a truly important step forward in principled unlearning and respectfully ask that you consider reevaluating your assessment in light of these clarifications. We remain eager to answer any additional questions you might have.

---

> > ### Comment · Reviewer_rC87 · 2025-08-07
> >
> > Thank you for the detailed rebuttal. It addresses most of my concerns. Regarding interpretability, the paper would benefit from incorporating lens-based analysis (e.g., Logit Lens) or n-gram-based methods (e.g., Jet n-grams). I have raised my score with the expectation that these additions will be included in the final version.

---

> > > ### Author Response · Authors · 2025-08-07
> > >
> > > We sincerely thank you for taking the time to review our work and for providing valuable feedback. Your comments have brought to light areas where the presentation of our paper can be improved, and we recognize that addressing these points will enhance the overall clarity and flow of the paper, as well as improve the delivery of the results.
> > >
> > > We will carefully revise the exposition of the paper to incorporate the thoughtful suggestions you have provided. Should there be any additional concerns or suggestions, we would be more than happy to address them. Thank you once again for your time and constructive remarks.

---

### Note · Authors · 2025-08-16

Dear Reviewers and AC,

We are grateful for this review process and for the different viewpoints brought up by the reviewers. We would like to offer this remark as a final summarization of our contributions.

In this paper, we formulated unlearning as a constrained optimization and solved it using primal-dual optimization. This primal-dual algorithm provides an optimal solution as established via strong duality.

Importantly, “The constrained optimization approach provides **explicit control over forgetting and retention, addressing limitations of scalarized loss methods.**“(P5bA) and “The primal-dual optimization framework provides a principled way to enforce retention constraints.” (W8eZ).
At the same time, the primal-dual formulation **does not incur any extra overhead and has the same exact computational complexity as all linear scalarization methods**.

Furthermore, we introduced a softmax-free Logit Flattening loss that drives the outputs towards uniformity on the forget set. “The logit-margin flattening loss is **numerically stable, softmax-free, and maintains non-vanishing gradients, enabling efficient optimization.**” (P5bA)
“Most unlearning methods are very sensitive to learning hyperparameters. The proposed method seems to be numerically stable. The proposed logit-margin loss is softmax-free, avoids vanishing gradients, and supports robust optimization for forgetting tasks.” (rC87).
We provided an extensive set of experiments evaluating our methodology. Due to space limitations, the majority of our experiments are deferred to the supplementary material. We provide **detailed analysis in the supplementary material** with
- radar charts and their areas as single metrics of unlearning success and utility,
- Single-turn jailbreak attempts
- Extra metrics (Membership Inference Attacks, Extraction Strengths, …)
- Longer unlearning
- etc

We conducted experiments on two established benchmarks for LLM unlearning: TOFU and MUSE. These benchmarks cover many challenging scenarios. Importantly, these benchmarks put the system under test for when there is significant overlap between the forget and retain sets and for when the forget set outnumbers the retain set. We also utilized a wide range of models, including Llama 2/3 1B-13B and Gemma 7B.

We are glad that we were able to address the concerns of the reviewers. We will more clearly state our contributions and clarify our exposition to avoid confusion in the camera-ready paper.

Best,

The Authors

---

### Decision · Program_Chairs · 2025-09-17

**Decision:**

Accept (poster)

**Comment:**

This paper uses a primal-dual optimization framework to address the unlearning problem in LLMs. The key advantage of the proposed method is the ability to explicitly control forgetting and retention in a principled way to enforce retention constraints. Some design choices of the proposed algorithm were also made to take into account practical considerations such as numerical stability. There are some limitations in this work that have been adequately discussed in the conclusion section. Overall, I believe that this is a significant contribution to LLM unlearning.

The rebuttal and subsequent discussions have addressed the reviewers' main concerns as well.